# Analysis of Etiology of Community-Acquired and Nosocomial Urinary Tract Infections and Antibiotic Resistance of Isolated Strains: Results of a 3-Year Surveillance (2020–2022) at the Pediatric Teaching Hospital in Warsaw

**DOI:** 10.3390/microorganisms11061438

**Published:** 2023-05-29

**Authors:** Monika Wanke-Rytt, Tomasz Sobierajski, Dominika Lachowicz, Dominika Seliga-Gąsior, Edyta Podsiadły

**Affiliations:** 1Department of Pediatrics with Clinical Assessment Unit, Medical University of Warsaw, 63a Zwirki i Wigury Str., 02-091 Warsaw, Poland; 2The Sociomedical Research Centre, Faculty of Applied Social Sciences and Resocialization, Warsaw University, 26/28 Krakowskie Przedmiescie Str., 00-927 Warsaw, Poland; tomasz.sobierajski@uw.edu.pl; 3Laboratory of Microbiology, University Center of Laboratory Medicine, 1a Banacha Str., 02-097 Warsaw, Poland; dominika.lachowicz@uckwum.pl (D.L.); edyta.podsiadly@uckwum.pl (E.P.); 4Department of Pharmaceutical Microbiology and Bioanalysis, Centre for Preclinical Research, Faculty of Pharmacy, Medical University of Warsaw, 1b Banacha Str., 02-097 Warsaw, Poland

**Keywords:** *Enterobacterales*, ESBL, pediatric, COVID-19, Poland, urinary tract infections, antibiotic resistance

## Abstract

Urinary tract infections (UTIs) remain the most common infections diagnosed in outpatients and hospitalized patients. This study was designed to determine the patterns of antibiotic resistance and the prevalence of uropathogens causing UTIs in pediatric patients hospitalized between 1 January 2020 and 31 December 2022 at Teaching Hospital in Warsaw. The most frequent species isolated from urine samples were *E. coli* (64.5%), *Klebsiella* spp. (11.6%), and *Enterococcus* spp. (6.1%). UTIs caused by *Enterobacter* spp., *Enterococcus* spp., and *Klebsiella* spp. were significantly more common in children younger than three months of age than in children older than three months (*p* < 0.001). Trimethoprim and trimethoprim–sulfamethoxazole were the least active compounds against *Enterobacterales* with the resistance of *E. coli*, *Klebsiella* spp., *P. mirabilis*, and *Enterobacter* spp. in the range of 26.7/25.2%, 48.4/40.4%, 51.1/40.4%, and 15.8/13.2% respectively. Ampicillin was also found to have resistance rates for *E. coli* of 54.9% and *P. mirabilis* of 44.7%. Cefalexin and cefuroxime were highly active towards *Enterobacterales* except for *Klebsiella* spp., in which the resistance level reached 40%. Regarding third- and fourth- generation cephalosporins, resistance in *E. coli* and *P. mirabilis* was observed in approximately 2–10% of the isolates, but in *Klebsiella* spp. and *Enterobacter* spp. ranged over 30%. The resistance of *Enterobacterales* to carbapenems, nitrofurantoin, and fosfomycin was below 1%. The quinolones resistance was very high for *Klebsiella* spp. (31.1%) and *P. mirabilis* (29.8%) and three times lower for *E. coli* (11.9%), *P. aeruginosa* (9.3%), *Enterobacter* spp. (2.6%), and *E. faecalis* (4.6%). Resistance to multiple antibiotic classes was identified in 396 *Enterobacterales* strains, 394 of which were multi-drug resistant (MDR) and 2 were extensive drug-resistant (XDR). In the case of *E. coli*, 30% of isolates were MDR, with the proportion of strains having this exact resistance pattern similar in all of the analyzed years; no *E. coli* XDR strains were isolated. The number of *Klebsiella* spp. MDR strains was much higher in 2022 (60%) than in 2021 (47.5%). In the analyzed time, only one strain of *K. pneumonia* XDR, producing New Delhi metallo-β-lactamase, was isolated. Monitoring infection trends is essential to improve control and limit the rise of bacterial resistance.

## 1. Introduction

Urinary tract infections (UTIs) are among the most common bacterial infections in the developmental age population [1]. Its occurrence in terms of younger children may indicate a urinary tract defect; the relationship between recurrent febrile UTIs and renal scarring or hypertension is still under investigation [2]. The guidelines of the American Academy of Pediatrics from 1999 emphasized the important role of UTIs in the pathogenesis of hypertension and chronic kidney disease [3]. More recent data, though, can only indicate a linkage between the occurrence of UTIs in terms of young children and impaired renal function in adulthood. Still, the strength of this relationship has yet to be determined [4,5].

The predominant organisms causing complicated and uncomplicated UTIs are uropathogenic *Escherichia coli*, *Klebsiella pneumoniae*, *Enterococcus faecalis*, *Proteus mirabilis*, and group B *Streptococcus* (GBS) [6]. Further, multi-drug-resistant *E. coli* and *K. pneumoniae* are continuously recognized to cause community-acquired UTIs (CA-UTIs) and hospital-acquired (HA) ones [7,8]. The rapid increase in the level of bacterial resistance to antibiotic use observed in recent years has made it necessary to revise views on the choice of antibiotic, the duration of antibiotic therapy, and the indications for chronic antibacterial prophylaxis. Moreover, there is a tendency to shorten the duration of antibiotic therapy and reduce antibacterial prophylaxis, which decrease the likelihood of selecting resistant strains.

Over the past years, the susceptibility profile of pathogens responsible for infections, including UTIs, has changed [9]. Resistance can vary over time depending on several factors, such as antibiotic use or changes in bacterial epidemiology. Therefore, data on pathogen susceptibility should be regularly updated [9].

*Enterobacterales*-producing extended-spectrum beta-lactamases (ESBL-PE) are resistant to most beta-lactams and often co-resistant to other classes of antibiotics [10,11], which limits therapeutic options [12], can lead to delays in an appropriate treatment, and increase the risk of complications [13]. According to the available literature, UTIs caused by ESBL-PE are associated with longer hospitalization and higher costs in adults [13]. In children, UTIs are among the most common infections caused by ESBL-PE [14,15].

According to the European Center for Disease Prevention and Control (ECDC) and the United States Centers for Disease Control and Prevention (CDC), MDR are strains that acquired non-susceptibility to at least one agent from three or more antimicrobial categories. Other profiles of acquired resistance are defined as extensive drug resistance (XDR), which is defined as a lack of susceptibility to at least one agent in all antimicrobial categories with the exception of a maximum of two. Finally, pandrug-resistance (PDR) is the lack of susceptibility to all agents in all categories of antimicrobials [16].

This study aimed to evaluate changes in the etiology of UTIs and the antibiotic susceptibility of isolated species in children hospitalized between 2020 and 2022.

## 2. Materials and Methods

### 2.1. Study Population

A retrospective cohort study was conducted from 1 January 2020 to 31 December 2022, at the Pediatric Teaching Hospital in Warsaw. The hospital is the biggest pediatric clinical center in central Poland, with 550 beds. The hospital treats patients from all over the country, demanding multi-specialty care in such departments as oncology, hematology, nephrology, neonatology, surgery, and intensive care. In addition, 24 h on-call services for general pediatric patients is provided.

A single isolate per patient collected from urine samples fulfilling the following criteria was included in the study: sample from the symptomatic patient (fever, dysuria, positive Goldflam sign, etc.), pyuria (≥10 white blood cell count/mm^3^), isolation of bacteria in urine culture in significant colony-forming units/mL (≥10^3^ in a urine sample collected by suprapubic puncture, ≥10^4^ in a urine sample collected by catheterization, ≥10^5^ in a urine sample collected from the midstream). In our hospital, a catheter-associated urinary tract infection (CAUTI) is diagnosed based on the presence of significant bacteriuria (≥10^5^) in patients with signs of fresh infection, as per the in-hospital recommendations. The infection control team representative is responsible for diagnosing CAUTI cases. Isolates considered as colonization were rejected. Both community-acquired and hospital-acquired cases of UTIs were enrolled. A total of 9084 clinical specimens were analyzed during the study period. For each isolate, the susceptibility to 17 antibiotics and chemotherapeutic were recorded. The age of the studied patients ranged from 1 day to 18 years.

### 2.2. Data Collection

From the laboratory documentation, patient information was collected including age, gender, species of cultured bacteria, and antibiotic sensitivity pattern of the etiological agent.

### 2.3. Microbiological Procedures

Bacterial identification was routinely performed with MALDI-TOF mass spectrometry with Microflex LT mass (Bruker, Mannheim, Germany) using the MBT Compass IVD software (Bruker Daltonics, Bremen, Germany) according to manufacturer’s instructions. Antimicrobial susceptibility testing (AST) was performed with the VITEK 2 instrument (BioMerieux, Durham, NC, USA) for the following antibiotics: ampicillin, piperacillin/tazobactam, cefuroxime, ceftazidime, cefotaxime, cefepime, meropenem, ertapenem, amikacin, gentamicin, tobramycin, ciprofloxacin, trimethoprim/sulfamethoxazole, nitrofurantoin and the Kirby–Bauer disk diffusion method for trimethoprim, cefixime, and fosfomycin. The results were interpreted in accordance with EUCAST guidelines according to the version applicable in a given year. Based on the pattern of antibiotic resistance, we classified the isolates as susceptible to all tested antibiotics (S), non-susceptible to at least one agent in three or more antimicrobial categories (MDR), or non-susceptible to at least one agent in all antimicrobial categories except two or less (XDR).

### 2.4. ESBL and Carbapenemase Detection

For isolates resistant to carbapenems, multiplex lateral flow immunoassay was used for the phenotypic detection and differentiation of five common carbapenemase families: *Klebsiella pneumoniae* carbapenemase (KPC), imipenemase (IMP), Verona integron encoded metallo-β-lactamase (VIM), New Delhi metallo-β-lactamase (NDM), and OXA-48. The test was performed according to the manufacturer’s instructions (Coris, BioConcept, Gembloux, Belgium). ESBL was detected with the phenotypic confirmation method—double-disk synergy test (DDST) according to EUCAST [17].

### 2.5. Statistical Analysis

Statistical analyses were performed in IBM SPSS Statistics 28.0.1.0. The relationship between variables was evaluated using the chi-squared test. For all analyses, a *p*-level of <0.05 was considered statistically significant.

## 3. Results

### 3.1. Etiology of UTI

Overall, 1393 clinical isolates from UTIs were obtained from 1 January 2020, to 31 December 2022. The recovered species are listed in Table 1.

The most common causative agent was *E. coli*, accounting for 58.6 to 70% of all cases depending on the year, followed by *Klebsiella* spp. 9.2–13.2%, and with a frequency of less than 5%, *Enterococcus* spp., *P. aeruginosa*, *P. mirabilis*, *Enterobacter* spp., and *Candida* spp. Over the three years of analysis, the share of *E. coli* in the total pool of infections decreased by 12 percentage points. The incidence of other agents increased by 2–3 percentage points, especially *Klebsiella* spp., but also, to lesser extent, *P. aeruginosa* and *Enterococcus* spp. There was a significant increase in UTIs caused by *Enterococcus* spp. in 2022 vs. 2020 (9.8% vs. 3.9%, *p* = 0.003).

Out of the 1393 culture-positive samples, 594 (42.6%) were from males and 799 (57.4%) were from females. In both sexes, *E. coli* was the most predominant species (girls—71.6%, boys—54.9%). In boys, *Klebsiella* spp. (14.1% in boys vs. 9.64% in girls), *P. mirabilis* (5.4% vs. 1.9%), *Enterobacter* spp. (5.2% vs. 1.9%), *Enterococcus* spp. (7.1% vs. 5.3%), and *P. aeruginosa* (5.1% vs. 3.0%) were isolated significantly more often (*p* < 0.001).

Based on age categories, the data contained specimens obtained from newborns and infants below three months of age (*n* = 304), and children over three months old (*n* = 1089) were analyzed. In both age groups, the dominant agent of UTIs was *E. coli.* UTIs caused by *Enterobacter* spp., *Enterococcus* spp., and *Klebsiella* spp., which were notably more common in newborns and babies below the third month of life compared to older ones (*p* < 0.001). From older children, substantially more often, *P. aeruginosa* and *P. mirabilis* were cultured (Figure 1).

### 3.2. Resistance Pattern of Uropathogens

#### 3.2.1. Resistance to Beta-Lactams

Acquired resistance to ampicillin was observed in 54.9% and 44.7% isolates of *E. coli* and *P. mirabilis*, respectively.

The highest prevalence of resistance to piperacillin/tazobactam was found in *Klebsiella* spp. strains (34.2%), with a large frequency also observed in *P. aeruginosa* (16.7%) and *E. coli* (5.5%). Such resistance was not ascertained in *P. mirabilis* and *Enterobacter* spp.

Cefalexin and cefuroxime were highly active towards *Enterobacterales* except for *Klebsiella* spp., where the resistance level reached 40%.

Regarding third- and fourth-generation cephalosporins, resistance in *E. coli* and *P. mirabilis* was observed in approximately 2–10% of the isolates. A higher prevalence of resistance was observed in *Klebsiella* spp. and *Enterobacter* spp. ranging over 30% of the isolated strains.

The resistance to carbapenems was inconsiderable in *Enterobacterales* and was below 1%, which consisted of one strain of *K. pneumoniae* producing NDM. It did not concern *Enterobacter* spp., in which 10.5% of strains were resistant to ertapenem. The resistance to carbapenems was much higher in the case of *P. aeruginosa* strains and involved 12% of the isolates; the resistance was associated with permeability, and no carbapenemases were detected (Table A1).

#### 3.2.2. Resistance to Other Classes of Antibiotics

Trimethoprim and trimethoprim–sulfamethoxazole were the least active compounds against *Enterobacterales*. The resistance of *E. coli*, *Klebsiella* spp., and *P. mirabilis* had varied values of 26.7/25.2%, 48.4/40.4%, and 51.1/40.4%, respectively. A much lower level of resistance to the chemotherapeutics was found in *Enterobacter* spp. (15.8/13.2%) and *E. faecalis* (4.6%/4.6%).

Aminoglycosides were highly active against all *Enterobacterales* species and *P. aeruginosa*, with an overall resistance rate of approximately 6%, except gentamycin, where the number of resistant strains varied between species: *Klebsiella* spp. (17.4%), *P. mirabilis* (8.5%), and *P. aeruginosa* (7.4%).

The rate *of E. coli* and *E. faecalis* isolates resistant to nitrofurantoin and fosfomycin was low, at around 1% for each molecule.

The quinolones resistance was very high for *Klebsiella* spp. (31.1%) and *P. mirabilis* (29.8%) and three times lower for *E. coli* (11.9%), *Enterobacter* spp. (2.6%), *E. faecalis* (4.6%), and *P. aeruginosa* (9.3%) (Table A1).

#### 3.2.3. Changes in Susceptibility in a Period of Three Years and the COVID-19 Pandemic in *E. coli* and *Klebsiella* spp.

There was a difference in beta-lactams resistance between 2020 and 2022. The *E. coli* isolates revealed a higher level of resistance to cephalosporins and piperacillin–tazobactam in strains isolated in 2020 compared to strains cultured in 2022 (*p* < 0.001). It is interesting to note that *Klebsiella* spp. strains that exhibited high levels of resistance in 2020 showed the lowest levels in the pandemic year of 2021, but then rapidly increased in the post-pandemic year of 2022. However, the percentage difference between 2020 and 2022 was not statistically significant, except for cefepime (*p* = 0.017).

The overall percentage of ESBL-producing *Enterobacterales* equaled 31.1% in *Klebsiella* spp., 9.8% in *E. coli*, 7.9% in *Enterobacter* spp., and up to 6.4% in *P. mirabilis.* The analysis of the difference in ESBL incidence between 2020 and 2022 showed a statistically significant increase in *Klebsiella* spp.—ESBL-positive strains— at 30% and 38% in 2020 and 2022, respectively (*p* < 0.001). No differences were observed in the number of *E. coli* producing ESBL.

Other tendencies observed in the three-year analysis encompassed a significant increase in resistance to cefepime in *Klebsiella* spp. (*p* = 0.017) and to trimethoprim and trimethoprim–sulfamethoxazole for *E. coli* (*p* = 0.019) and for *Klebsiella* spp. trimethoprim (*p* < 0.01) and trimethoprim–sulfamethoxazole (*p* = 0.021).

No statistically significant differences were detected for the other antibiotics (Table A2).

#### 3.2.4. Resistance Profile—S, MDR, XDR, PDR

Regarding the pattern of resistance, we found that 409 out of 1393 isolates were resistant to multiple antibiotics with MDR and XDR profiles. A resistance to different classes of antibiotics has been found in 396 strains of *Enterobacterales*, of which 394 were MDR, and 2 were XDR. An MDR resistance pattern was detected in 12 out of 54 *P. aeruginosa* strains and 1 out of 65 *Enterococcus* spp. isolates.

For *E. coli*, the proportion of MDR strains was similar in each analyzed year, equaling around 30%. The number of *Klebsiella* spp. MDR strains was notably higher: in 2022, it equaled 60%, and it was distinctly lower in 2021 (47.4%). The percentage of MDR *P. mirabilis* and *Enterobacter* spp. strains was 29.8 and 21.1%, respectively. No XDR strains of *E. coli* and *Enterobacter* spp. were isolated in the analyzed period. One isolate each of *P. mirabilis*, *K. pneumoniae*, and *P. aeruginosa* presented an XDR profile.

The number of fully susceptible strains in *Enterobacterales* ranged from 2.5% to 35.1%, depending on the species and year of analysis. The highest percentage of such strains was found among *E. coli* isolates: 13.4%, 35.1%, and 34.1%, respectively, in the consecutive years of 2020–2022. In terms of *Klebsiella* spp.: 2.5%, 5.1%, and 12.9% were fully susceptible in the following years. Concerning other *Enterobacterales*, 23.4% *P. mirabilis* and 18.4% *Enterobacter* spp. had this profile. No PDR strains were detected (Figure 2).

## 4. Discussion

Our study presents the distribution of etiology and antibiotic susceptibility patterns of microbial species isolated from patients with community and nosocomial UTIs between January 2020 and December 2022 at the Pediatric Teaching Hospital in Poland. As expected, as in other studies, *E. coli* was the leading species recovered from UTIs in children [18]. Stefaniuk et al., in a multicenter analysis conducted on the Polish population in 2013, found that *E. coli* was responsible for 71.4% of all UTIs [19]. Other Gram-negative etiological agents of UTIs confirmed in our study were *Klebsiella* spp., *P. aeruginosa*, *Enterobacter* spp., and *Proteus* spp., with a notable contribution of *Klebsiella* spp. accounting for 11.6% of the isolates. The Gram-positive human uropathogens identified in our study included *Enterococcus* spp. and *Staphylococcus* spp.

The analysis of samples from children under 3 months of age showed a considerable higher incidence of etiology of *Enterobacter* spp., *Enterococcus* spp., and *Klebsiella* spp. than in older children. A study by Kot et al. showed that *K. pneumoniae* was isolated more often in patients over 60 years of age than in children [20]. This can be related to the reduced efficiency of the immune system and impaired function of the urinary tract in the case of patients in extreme age groups, which may explain our result of the group of similar immunological status. In our analysis, unlike other publications, we did not show an important contribution of *S. agalactiae* to the etiology of UTI in newborns, which may be related to the screening of women for *S. agalactiae* colonization and procedures in place for perinatal antibiotic therapy [21].

The global spread of MDR microorganisms has led to an increase in the incidence of difficult-to-treat UTIs, a phenomenon also observed in children [22]. The emergence of multi-resistant uropathogens makes the treatment of these infections challenging. Mahony et al., in their review, emphasize that MDR infections are increasingly common in individuals with no history of hospitalization or prior hospitalization [22]. What also needs to be noted is that MDR infections can be both community-acquired and hospital-acquired [22]. Our results differ slightly from those of Miftode et al., who obtained MDR rates among *E. coli*, *Klebsiella* spp., and *Proteus* spp. of 60.3, 18.2, and 15.8%, respectively [23]. Our study detected half as many MDR *E. coli* strains (30%), but more MDR *Klebsiella* spp. (60%) and a similar level of MDR *Proteus* spp. (14%). The data presented in the paper by Miftode et al. were from 2019, which might suggest the presence of an increased resistance of *K. pneumoniae*, causing UTIs.

The increasing prevalence of ESBL strains in pediatrics may result in more frequent failures of empiric therapy. In our study, we found no increase in the prevalence of *E. coli* producing ESBL. However, we noted an increase in the incidence of UTI caused by ESBL -producing *Klebsiella* spp. Moreover, compared to the French data [24], we found a much higher percentage of ESBL-producing strains. Farfour et al. reported a rate of ESBL-producing strains in the pediatric ward of 4.5%, while in our study, it ranges from 4.6% in *P. mirabilis* up to 31.1% in *Klebsiella* spp. The observation may indicate an increase in the number of ESBL-producing strains in UTIs in terms of children in the last years [24]. For pyelonephritis and extra-urinary tract infections (including urosepsis) caused by ESBL-producing strains, carbapenems are a treatment of choice. In our study, we did not show resistance to carbapenems of pathogens responsible for UTIs. In recent years, carbapenem resistance due to the production of carbapenemases has been increasingly reported among clinical isolates worldwide [25].

Considering the risk of kidney damage in children with delayed initiation of appropriate treatment, the selection of a suitable empirical therapy is crucial. According to Polish recommendations [26], cefuroxime is the first-line drug in UTIs due to its good renal penetration and good susceptibility profile. In our study, *E. coli*’s resistance to cefuroxime did not exceed 20% throughout the study period. Unlike *E. coli*, *Klebsiella* spp. was resistant in almost half of the cases. In another Polish study, the results of *E. coli* susceptibility to a second-generation cephalosporin showed about 30% of resistant strains, and in the case of *K. pneumoniae,* as much as 80% [22]. In some countries, such as Turkey, the resistance of *E. coli* and *K. pneumoniae* to cefuroxime reaches 80%, eliminating this antibiotic from empirical therapy [27].

A worrying trend in our results is the increasing resistance of *Klebsiella* spp. and its increasing occurrence as an etiological agent of UTIs. The higher resistance of *Klebsiella* spp. to beta-lactam antibiotics in 2020 than in the full pandemic year of 2021 may be due to hospital-acquired infections, and their lower level in 2021 may be a result of fewer hospitalizations. An increase in terms of resistance in the post-pandemic year 2022 may be derived from the influx of refugees from Ukraine and the medical services provided to them. The increase in resistance is associated with selecting strains resistant to cephalosporins and fluoroquinolones. Fluoroquinolones are primarily reserved for adults. However, they also found their place in pediatrics due to protocols for the treatment in oncology and gastroenterology. The use of fluoroquinolones in children remains low, but some centers are reporting an increase in their usage [28,29,30]. The data suggest that bacterial isolates from the urinary tract of individual children who received prior antibiotic prescriptions were more likely to be resistant to antibiotics. This increased risk could persist for up to six months [31]. The recent use of antibiotics, including treatment and prophylaxis, is considered a risk factor for MDR UTIs, although the duration of exposure varies across studies. A study by Raman et al. found that any antibiotic use in children in the past month, including antibiotic prophylaxis, was associated with an increased risk of *E. coli* MDR UTIs [32].

According to the recommendations of Poland and the European Union for UTIs, ampicillin, trimethoprim, and trimethoprim–sulfamethoxazole should not be used for empirical treatment due to the high percentage of resistance, which was confirmed in our analysis in which we found high resistance to all of them. In the systematic review and meta-analysis of Bryce et al., the authors indicated that in countries belonging to the Organization for Economic Co-operation and Development (OECD), the cumulative prevalence of resistance to *E. coli* was 53.4% for ampicillin, 23.6% for trimethoprim, 8.2% for co-amoxiclav, and 2.1% for ciprofloxacin; nitrofurantoin was the lowest at 1.3%. The resistance in studies in non-OECD countries was notably higher: 79.8% for ampicillin, 60.3% for co-amoxiclav, 26.8% for ciprofloxacin, and 17.0% for nitrofurantoin [31]. Since February 2022, due to the war in Ukraine (outside the OECD), we have observed a considerable increase in the number of refugee patients in our hospital, which is also associated with the increased isolation of MDR strains. Despite its clinical and epidemiological significance, few studies have been conducted on the prevalence of MDR in Ukrainian hospitals [33,34]. Data from the ECDC indicate a high prevalence of an invasive MDR Gram-negative bacteria in Ukraine in 2020, where almost 77% of *Acinetobacter baumannii* and 84% of *K. pneumoniae* were found to be carbapenem-resistant [35]. Similar data were presented by Schultze et al., who showed a similar carbapenem-resistant *Enterobacterales* (CRE) profile of pathogens in the case of Ukrainian patients who required treatment in a German hospital [36].

An additional problem specific to Poland is the assessment of furazidine, which is available in our country without prescription. The problem is the inability to determine drug susceptibility. There are no tests for the in vitro diagnosis of susceptibility to this chemotherapeutic. The results are unreasonably extrapolated from the determination of nitrofurantoin, which is unavailable on the Polish market. Nitrofurantoin, and in the case of Poland, furazidine, is recommended in the treatment (and prophylaxis) of uncomplicated cystitis as empirical therapy. Previous studies conducted in Romania, Poland, and France showed that the percentage of *E. coli* resistant to nitrofurantoin ranged from 3 to 3.8%, which is similar to our findings [37,38,39]. However, there are still alarming reports of a high percentage of strains resistant to nitrofurantoin, even reaching 20%, which should prompt caution in the use of furazidine in the treatment of UTIs [9,10].

## 5. Conclusions

Our analysis of strains cultured from the urine of pediatric patients hospitalized for UTIs confirmed that the main etiologic agent in this group of patients is *E. coli*, with *Klebsiella* spp. also frequently isolated. The results also showed that *Enterobacter* spp., *Enterococcus* spp., and *Klebsiella* spp. were isolated more frequently in children under 3 months of age than in older children. This information can be considered when prescribing treatment for urinary tract infections. We observed no significant changes in sensitivity to beta-lactams for *E. coli* and *Klebsiella* spp., so current treatment recommendations for UTIs in children at our hospital remain the same. We also noted that *Klebsiella* spp. and *Enterobacter* spp. are increasingly being isolated from the urine of younger patients, which is a worrisome trend in pediatrics and should be closely monitored.

## Figures and Tables

**Figure 1 microorganisms-11-01438-f001:**
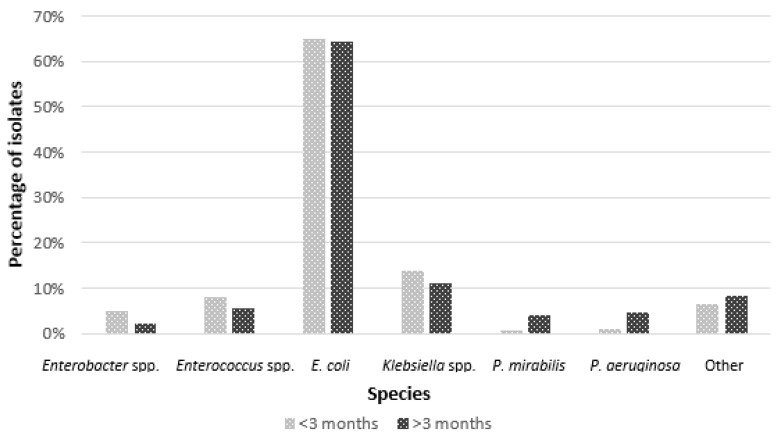
Distribution of etiology of UTIs according to age (*p* < 0.001).

**Figure 2 microorganisms-11-01438-f002:**
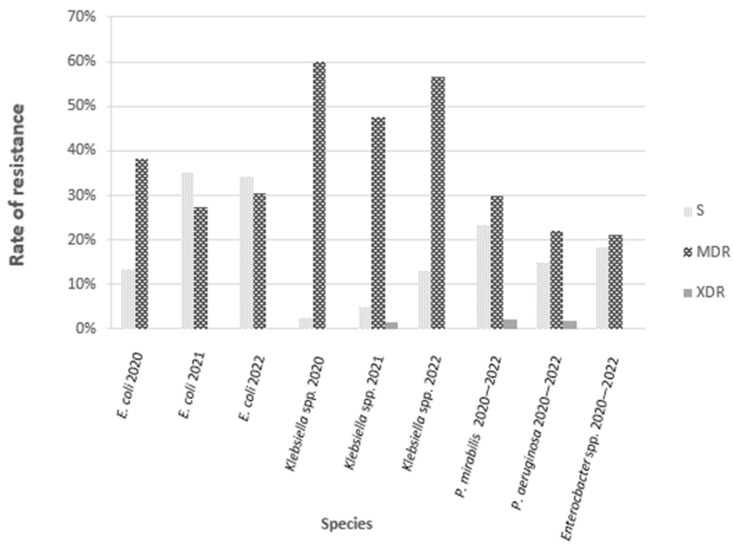
Distribution of susceptible (S), multidrug-resistant (MDR), and extensive drug-resistant (XDR) profiles among strains isolated from UTIs in children.

**Table 1 microorganisms-11-01438-t001:** Distribution of bacterial species isolated from UTIs in pediatric population during the three-year study period.

Etiological Agent	2020		2021		2022		Total
	N	%	N	%	N	%	%
*Escherichia coli*	306	70.0	316	65.2	276	58.6	64.5
*Klebsiella* spp. (*K. pneumoniae, K. oxytoca, K. variicola*)	40	9.2	59	12.2	62	13.2	11.6
*Enterococcus faecalis*	15	3.4	18	3.7	32	6.8	4.7
*Pseudomonas aeruginosa*	14	3.2	18	3.7	22	4.7	3.9
*Proteus mirabilis*	19	4.3	12	2.5	16	3.4	3.4
*Enterobacter* spp.	9	2.1	14	2.9	15	3.2	2.7
*Candida* spp.	10	2.3	12	2.5	8	1.7	2.2
*Enterococcus faecium*	2	0.5	3	0.6	14	3.0	1.4
*Citrobacter* spp.	3	0.7	5	1.0	3	0.6	0.8
*Serratia marcescens*	4	0.9	3	0.6	2	0.4	0.6
*Klebsiella aerogenes* (*Enterobacter aerogenes*)	3	0.7	4	0.8	1	0.2	0.6
*Staphylococcus saprophyticus*	0	0.0	4	0.8	3	0.6	0.5
*Streptococcus agalactiae*	0	0.0	3	0.6	4	0.8	0.5
*Morganella morganii*	2	0.5	2	0.4	1	0.2	0.4
*Corynebacterium* spp.	2	0.5	0	0.0	2	0.4	0.3
*Aerococcus urinae*	0	0.0	0	0.0	4	0.8	0.3
Other	8	1.8	12	2.5	6	1.3	1.9
TOTAL	437		485		471		1393

## Data Availability

The data presented in this study are available in the article.

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
