# Peer review of "Analysis of Etiology of Community-Acquired and Nosocomial Urinary Tract Infections and Antibiotic Resistance of Isolated Strains: Results of a 3-Year Surveillance (2020–2022) at the Pediatric Teaching Hospital in Warsaw"

_microorganisms, 2023, doi:10.3390/microorganisms11061438_

Round 1

Reviewer 1 Report

Monika Wanke-Rytt et al. studied the distribution of etiology and antibiotic susceptibility patterns of microbial species isolated from patients with community and hospital-acquired UTIs between January 2020 and December 2022 in the Pediatric Teaching Hospital in Poland. This is a comprehensive study with a cohort of significance that could help to understand the prevalence of antibiotic resistance in UTIs in that community with treatment implications. However, there are several concerns that should be solved before acceptance for publication.

In general, authors refer to species casually and occasionally, what causes poor reading comprehension. In addition, they do not say at all what Table they are talking about. Please, read the manuscript throughout and check it. 

1- Line 32: “The number of K. pneumoniae MDR strains was much higher and  equaled 60 %, it was distinctly lower in 2021 (47,5%).” you should write down what you are comparing to? 

2- Line 130: “followed by Klebsiella 9.2 – 13.2%, and with frequency lower than” Please, add sp

3- Line  134: “especially Klebsiella but also, to less extent, Pseudomonas and Enterococcus.” Please add the species

4- Lines  134- 135: “In 2022, there was a significant increase in urinary tract infections caused by Enterococcus spp.” Please, clarify which values from Table 1 you are considering to do this statement, as well as the p value and statistics carried out.

5- Lines  138-140: “In boys, significantly more often, Klebsiella spp. (14.1% in boys vs. 9.64% in girls), Proteus mirabilis (5.4% vs. 1.9%), Enterobacter spp. (5.2% vs. 1.9%), Enterococcus spp. (7.1% vs. 5.3%) and P. aeruginosa (5.1% vs. 3.0%) were isolated.” Please, could you add  the p value and statistics that support this statement?

6- Lines 141-149: “Based on age categories, the data contained specimens obtained from newborns and infants below three months of age (n = 304), and children over three months old (n=1089) were analyzed. In both age groups, the dominant factor of UTI is E. coli. Neonates and infants below three months of age comprise a high proportion of UTIs caused by Entero- coccus, Enterobacter, and Klebsiella isolation compared to older children. However, from older children, Pseudomonas aeruginosa and Proteus mirabilis were cultured significantly more often. Urinary tract infections caused by Enterobacter, Enterococcus, and Klebsiella spp were significantly more common in children < 3 months than in children > 3 months (p< 0.001”. It would be clarifying to show these results on a table or a graph with actual data and also to put statistics on it.

7- Line 150- 201: “3.2. Resistance pattern of uropathogens”. Authors should help readers by adding the reference to the Table you are talking about.

8- Line 152: “Acquired resistance to amoxicillin was observed in 54.9% and 44.7% of E. coli” Are you talking about Table 2? Did you mean ampicillin?

9- Line 155: “Klebsiella spp. strains (34.2%), it was also with high frequency observed in P. aeruginosa” Did you mean Klebsiella sp.? 

10- Lines 185-186: “A difference was seen in resistance to beta-lactams when compared between 2020 and 2022. E. coli isolates revealed a higher resistance level to cephalosporins and piperacillin-tazobactam in strains isolated in 2020 compared to strains cultured in 2021 and 2022.” Authors should write down statistics that support this statement or refer to differences as “statistically non-significant trend”.

11- Lines 191-220 : It would improve understanding and visualization of results to show XDR,MDR and ESBL results on a table or a graph.

12- Line 233: “higher prevalence of etiology of Enterobacter, Enterococcus, and Klebsiella spp. then in older”Please, replace “then” for “than”.

13- Lines 238-239: “In our analysis, unlike other publications, we did not show a significant share of S. agalaciae in etiology of UTI in newborns [21].” Could you explain the implications of this finding?

14- Lines 242-243: “The authors of the review emphasize that MDR infections are increasingly common in…” It is not clear what are you referring to

15- Lines 275-278: “Higher resistance of K. pneumoniae to beta-lactam antibiotics in 2020 than in the full pandemic year 2021 might result from acquired hospital infections, and their lower level in 2021 might result from a reduced number of hospitalizations”. This paragraph is not supported by data shown, given that you did not observe statistically significant differences on these results, as stated in lines 189-190: “However, percentage between years were not statistically significant.”   

16- Lines 278-311: Authors argue that the implications of the influx of refugees from Ukraine in the increased antibiotics resistance since 2022. However, the only demonstrated one K. pneumoniae NDM in a refugee patient form Ukraine (Lines 310-311). I think this is not enough evidence to arrive at that conclusion.

17- Table 2: There is no reference for LEV on the abbreviations used and GM is shown twice with disagreeing results

18- Supporting information is not available on the link provided

Author Response

Monika Wanke-Rytt et al. studied the distribution of etiology and antibiotic susceptibility patterns of microbial species isolated from patients with community and hospital-acquired UTIs between January 2020 and December 2022 in the Pediatric Teaching Hospital in Poland. This is a comprehensive study with a cohort of significance that could help to understand the prevalence of antibiotic resistance in UTIs in that community with treatment implications. However, there are several concerns that should be solved before acceptance for publication.

In general, authors refer to species casually and occasionally, what causes poor reading comprehension. In addition, they do not say at all what Table they are talking about. Please, read the manuscript throughout and check it. 

  • Line 32: “The number of K. pneumoniae MDR strains was much higher and  equaled 60 %, it was distinctly lower in 2021 (47,5%).” you should write down what you are comparing to? 

Thank you for noticing, the currently corrected comparison can be seen in line 47. 

  • Line 130: “followed by Klebsiella 9.2 – 13.2%, and with frequency lower than” Please, add sp

We have added, the change seen in line 155

  • Line  134: “especially Klebsiella but also, to less extent, Pseudomonas and Enterococcus.” Please add the species

We have added, the change seen in line 159

  • Lines  134- 135: “In 2022, there was a significant increase in urinary tract infections caused by Enterococcus spp.” Please, clarify which values from Table 1 you are considering to do this statement, as well as the p value and statistics carried out.

Thank you. We calculated the significance (p= 0.003) which is added and can be seen in line 160-161. We took into account E. faecalis and E. faecium 

  • Lines  138-140: “In boys, significantly more often, Klebsiella spp. (14.1% in boys vs. 9.64% in girls), Proteus mirabilis (5.4% vs. 1.9%), Enterobacter spp. (5.2% vs. 1.9%), Enterococcus spp. (7.1% vs. 5.3%) and P. aeruginosa (5.1% vs. 3.0%) were isolated.” Please, could you add the p value and statistics that support this statement?

We've added p value what is shown in line 166 

6- Lines 141-149: “Based on age categories, the data contained specimens obtained from newborns and infants below three months of age (n = 304), and children over three months old (n=1089) were analyzed. In both age groups, the dominant factor of UTI is E. coli. Neonates and infants below three months of age comprise a high proportion of UTIs caused by Entero- coccus, Enterobacter, and Klebsiella isolation compared to older children. However, from older children, Pseudomonas aeruginosa and Proteus mirabilis were cultured significantly more often. Urinary tract infections caused by Enterobacter, Enterococcus, and Klebsiella spp were significantly more common in children < 3 months than in children > 3 months (p< 0.001”. It would be clarifying to show these results on a table or a graph with actual data and also to put statistics on it.

Thank you for noticing, I agree that the graphic form will be more readable. Therefore, we have added Figure 1presenting visually the etiologies in relation to age. 

  • Line 150- 201: “3.2. Resistance pattern of uropathogens”. Authors should help readers by adding the reference to the Table you are talking about.

Of course, this is our oversight. We added a reference to what is visible in line 196

  • Line 152: “Acquired resistance to amoxicillin was observed in 54.9% and 44.7% of E. coli” Are you talking about Table 2? Did you mean ampicillin?

Yes, of course. The change shown in line 181 

  • Line 155: “Klebsiella spp. strains (34.2%), it was also with high frequency observed in P. aeruginosa” Did you mean Klebsiella sp.? 

We have changed the sentence formation to make it clearer now - line 184-185 

  • Lines 185-186: “A difference was seen in resistance to beta-lactams when compared between 2020 and 2022. E. coli isolates revealed a higher resistance level to cephalosporins and piperacillin-tazobactam in strains isolated in 2020 compared to strains cultured in 2021 and 2022.” Authors should write down statistics that support this statement or refer to differences as “statistically non-significant trend”.

We calculated and added statistical significance to the text, which is currently visible in line 218. 

  • Lines 191-220 : It would improve understanding and visualization of results to show XDR,MDR and ESBL results on a table or a graph.

We have introduced Figure 2, which shows MDR and XDR, for greater readability.

  • Line 233: “higher prevalence of etiology of Enterobacter, Enterococcus, and Klebsiella spp. then in older”Please, replace “then” for “than”.

We have made a change -line 268 

13- Lines 238-239: “In our analysis, unlike other publications, we did not show a significant share of S. agalaciae in etiology of UTI in newborns [21].” Could you explain the implications of this finding?

We introduced an explanation (hypothesis) in the text - line 274-276 

14- Lines 242-243: “The authors of the review emphasize that MDR infections are increasingly common in…” It is not clear what are you referring to

We changed the sentence formation and added the author and references to make it readable - line 280-281 

15- Lines 275-278: “Higher resistance of K. pneumoniae to beta-lactam antibiotics in 2020 than in the full pandemic year 2021 might result from acquired hospital infections, and their lower level in 2021 might result from a reduced number of hospitalizations”. This paragraph is not supported by data shown, given that you did not observe statistically significant differences on these results, as stated in lines 189-190: “However, percentage between years were not statistically significant.”   

Thank you for this comment. As it is mentioned in the text in 2021 resistance was rarer than in 2020 however in 2022 we noticed a worrying trend of increase, which at this point is not significant statistical

16- Lines 278-311: Authors argue that the implications of the influx of refugees from Ukraine in the increased antibiotics resistance since 2022. However, the only demonstrated one K. pneumoniae NDM in a refugee patient form Ukraine (Lines 310-311). I think this is not enough evidence to arrive at that conclusion.

Yes, I completely agree with this comment. In the revised manuscript, we did not include this statement, which actually did not explain the thread under discussion.

17- Table 2: There is no reference for LEV on the abbreviations used and GM is shown twice with disagreeing results

The errors that were highlighted have been rectified in the table.

18- Supporting information is not available on the link provided

Reviewer 2 Report

The manuscript is written neatly. however, minor revision  is required

1-The keywords are not expressive, please change them

2- In the introduction section lines 76 and 77, please clarify the aim and mention more details

3-In the conclusion section, please mention more details

Author Response

The manuscript is written neatly. however, minor revision  is required

1-The keywords are not expressive, please change them

Thank you for noticing, we have made the corrections as suggested.

2- In the introduction section lines 76 and 77, please clarify the aim and mention more details

Thank you, we have detailed the aim of the study

3-In the conclusion section, please mention more details

Thank you, we have detailed this section

Reviewer 3 Report

I have read with interest the manuscript submitted by Wanke-Rytt et al.

I have some comments to be addressed in order to improve the quality of the manuscript:

- all over the manuscript, the full name of the bacteria should be utilized only at the first use, further mentions should include only the short form.

- once you define an abbreviation, such as UTI, further mentions should utilize only the abbreviation.

- bacterial names followed by spp should be written "spp.

- rows 57-58 - provide the reference for this information.

- only the first use in the text of a term should include the abbreviation (for example multidrug-resistance - row 49-71)

Materials and methods: please define clinical syndrome and the values for the CFU/mL

Results

- Table 1 - bacterial name should be italicized; please consider also adding the total number of each bacteria, not only the percentage.

- row 135 - was a significant increase in urinary tract infections UTIs caused by Enterococcus Enterococcus spp.

- how do the authors support the word "significant" (for ex rows 135, 138, 190, etc)? please provide more advanced statistics, other than percentages.

- there is chaotic writing of bacterial names all over the document, both correct and incorrect forms even in the same sentence.

-I highly recommend adding supplementary tables and figures.

- I would have expected to find some information regarding the clinical presentation, with differences between higher and lower UTIs. 

- Can you provide any information about the hospitalization length/mortality and the relationship between the antibiotic resistance profiles and the outcome of the patients?

- it seems to be an additional table at the end of the manuscript.

Author Response

I have read with interest the manuscript submitted by Wanke-Rytt et al.

I have some comments to be addressed in order to improve the quality of the manuscript:

- all over the manuscript, the full name of the bacteria should be utilized only at the first use, further mentions should include only the short form.

Thank you for noticing, we have made the corrections as suggested.

- once you define an abbreviation, such as UTI, further mentions should utilize only the abbreviation.

I agree, we tried to convert to an abbrevation in places where it was appropriate.

- bacterial names followed by spp should be written "spp.

Thank you for noticing, we have made the corrections as suggested.

- rows 57-58 - provide the reference for this information.

We have changed the sentence formation and referred to better literaturÄ™

- only the first use in the text of a term should include the abbreviation (for example multidrug-resistance - row 49-71)

We have made the corrections as suggested

Materials and methods: please define clinical syndrome and the values for the CFU/mL

Thank you for noticing, we have made the corrections as suggested.

Results

- Table 1 - bacterial name should be italicized; please consider also adding the total number of each bacteria, not only the percentage.

Thank you for noticing, we have made the corrections as suggested.

- row 135 - was a significant increase in urinary tract infections UTIs caused by EnterococcusEnterococcus spp.

Thank you for noticing, we have made the corrections as suggested -  row  60

- how do the authors support the word "significant" (for ex rows 135, 138, 190, etc)? please provide more advanced statistics, other than percentages.

Thank you for your insightful input. We have conducted a statistical analysis and incorporated it into the text to ensure accuracy.

- there is chaotic writing of bacterial names all over the document, both correct and incorrect forms even in the same sentence.

Thank you for the right comment. As suggested, we have cleaned up the nomenclature.

-I highly recommend adding supplementary tables and figures.

As suggested, we added Figures 1 and 2 making the results more readable

- I would have expected to find some information regarding the clinical presentation, with differences between higher and lower UTIs. 

In this retrospective study, we focused primarily on the microbiological aspect. Wanting to focus on the clinical picture as well, it would definitely be better to conduct a prospective study to avoid methodological errors

- Can you provide any information about the hospitalization length/mortality and the relationship between the antibiotic resistance profiles and the outcome of the patients?

Although it would enhance the study's value, a prospective study would be more suitable to assess these correlations. Our study primarily concentrated on the microbiological aspect.

- it seems to be an additional table at the end of the manuscript.

Thank you for your suggestion. We have added it as requested.

Round 2

Reviewer 1 Report

All my requests were solved

Author Response

Thank you. 

Kind regards. 

Reviewer 3 Report

I appreciate the authors' efforts in editing the manuscript, the English language is now much better. 

I still consider that more elaborate statistics would significantly increase the manuscript's value. Moreover, in the results section, when the authors mention that a difference was not statistically significant, the p-value should be included to support the affirmation.

The cut-off value of 103 CFU/mL was used for all patients? This might lead to an extremely important over-diagnosis of UTIs.

How do you define the clinical presentation for urinary catheterized patients, since you mentioned that you included only symptomatic patients? This mention should be included in the Materials and Methods section.

Author Response

I still consider that more elaborate statistics would significantly increase the manuscript's value. Moreover, in the results section, when the authors mention that a difference was not statistically significant, the p-value should be included to support the affirmation.

Authors’ response: I appreciate you bringing this to our attention. It seems that the paragraph may not be entirely clear, so we made some adjustments to improve its clarity. Additionally, we have calculated the irrelevance and would like to note that while it is present, we have chosen not to include it in the text as we believe it would not be interesting to a reader.

CROX - p = 0.584

CFN - p = 0.418

CTX - p = 0.620

CROPO - p = 0.510

TZP - p = 0.201

CFX - p = 0.634

TAZ – p =

The cut-off value of 103 CFU/mL was used for all patients? This might lead to an extremely important over-diagnosis of UTIs.

How do you define the clinical presentation for urinary catheterized patients, since you mentioned that you included only symptomatic patients? This mention should be included in the Materials and Methods section.

Authors’ response: Thank you for bringing this detail to our attention. We have provided a detailed explanation of the CFU criteria adopted for each patient. Additionally, we have included an explanation of CAUTI infections, which can be found in lines 109-112.
